# Planispine A Sensitized Cancer Cells to Cisplatin by Inhibiting the Fanconi Anemia Pathway

**DOI:** 10.3390/molecules27217288

**Published:** 2022-10-26

**Authors:** Thangjam Davis Singh, Ningthoujam Indrajit Singh, Khuraijam Mrinalini Devi, Remmei Meiguilungpou, Lhaineichong Khongsai, Lisam Shanjukumar Singh, Naresh Chandra Bal, Ningombam Swapana, Chingakham Brajakishor Singh, Thiyam Ramsing Singh

**Affiliations:** 1Department of Biotechnology, Manipur University, Canchipur, Imphal 795003, Manipur, India; 2Institute of Bioresources and Sustainable Development, Takyelpat, Imphal 795001, Manipur, India; 3School of Biotechnology, Kalinga Institute of Industrial Technology (KIIT), Bhubaneswar 751024, Odisha, India; 4Department of Chemistry, Manipur Technical University, Imphal 795001, Manipur, India; 5Department of Molecular Biology and Biotechnology, Tezpur University, Napaam, Tezpur 784028, Assam, India

**Keywords:** *Zanthoxylum*, cisplatin, apoptosis, FANCD2, chemotherapy

## Abstract

The use of cisplatin as a chemotherapeutic drug is impeded by the development of drug resistance. Combination therapies of a chemosensitizer for cisplatin have been studied, but with little success, and the search for an effective combination therapy is continuing. Our earlier reports have shown that *Zanthoxylum armatum* DC. extract enhances the apoptotic effect of cisplatin in cancer cell lines. In this study, we purified and identified the bioactive phytocompound through bio-assay-guided purification, using column chromatography and HPLC. Chemical characterization using NMR and mass spectrometry revealed the compound as planispine A, with molecular structure C_25_H_30_O_6_ and molecular weight, 426.16 g/mol. Planispine A was found to inhibit cancer cell proliferation in a dose-dependent manner and to sensitize the cancer cells to cisplatin-augmented apoptotic cell death, in a caspase-dependent manner. A combination of planispine A and cisplatin induced S-phase cell cycle arrest, and reduced the expression of survival proteins such as cyclin D1. Interestingly, planispine A inhibits the Fanconi anemia pathway, as shown by reduced FANCD2 foci formation and FANCD2 monoubiquitination, which revealed the molecular mechanism of chemo-sensitization of cancer cells to cisplatin. Evaluation of this combination therapy in cisplatin-resistant tumors may lead to more efficient cisplatin treatment.

## 1. Introduction

The currently available chemotherapeutic drug, cisplatin, is very effective; it specifically eliminates most cancer cells initially at early stage of treatment, but eventually relapses with cisplatin-resistant tumors, causing genetic and physiological side effects [1,2,3]. Over time, drug resistance arises through enhanced DNA damage repair, genetic response, metabolic effects, growth factors and drug efflux with altered membrane transport [4,5,6]. 

To overcome the cisplatin-resistant cancer, certain strategies need to be employed, such as finding novel cytotoxic phytocompounds, the modulation of apoptosis, and drug combinations to develop new treatment strategies [7,8,9]. Medicinal plants provide important clues for identifying and developing synergistic drugs that reduce side effects and toxicity, and enhance the therapeutic potential of the chemotherapeutic drug, cisplatin [10,11].

The Fanconi anemia (FA) pathway is involved during the repair of DNA interstrand crosslink (ICL), and regulates the cellular response to cisplatin [12,13]. We are exposed to various sources of exogenous (chemotherapeutic drugs–mitomycin C, cis-platinum, etc.) and endogenous (aldehydes) ICL-inducing agents, from food habits or the environment [14]. Most of this damage is repaired through the FA/BRCA pathway, and any defect in this pathway results in chromosomal instability, sensitivity to DNA cross-linking agents, and increased risk of cancer, bone marrow failure and developmental abnormalities [15,16].

Since the Fanconi anemia pathway is involved in the repair of ICL, the mutation of FA gene(s) or the disruption of the FA pathway leads to hypersensitivity of cancer cells to ICL drugs such as cisplatin or MMC (Mitomycin C) [17,18]. Hence, inhibitors of the FA pathway may be used as chemo-sensitizers for cross-linking chemotherapeutic agents (e.g., cisplatin) in cancer treatment [19]. It is evident that a better understanding of this complex pathway remains a priority for medical research, and would lead to the discovery of ways to make cancer chemotherapy more effective and with greater therapeutic window for overcoming chemo drug-induced toxicity and drug resistance.

Screening and purification of plant metabolites would yield new molecules that disturbed the FA pathway, thereby sensitizing the cancer cells to ICL drugs [20,21]. Our previous work has d reported that *Z. armatum* leaf extract shows a synergistic effect on the anti-cancer activity of the chemotherapeutic drug, cisplatin [22]. In the present study, we purified the bioactive compound from the leaf extract of *Z. armatum* through bioassay-guided purification, and identified planispine A (lignans), which can act as a chemo sanitizer, possibly by disrupting the FA pathway.

## 2. Results

### 2.1. Characterisation of Bioactive Compounds

*Z. armatum* DC. leaf extracts were found to induce apoptosis and enhance the therapeutic potential of cisplatin [22].Therefore, we wanted to purify the bioactive compound using the bioactive-guided fractionation protocol, and the general flowchart for the purification is shown below (Figure 1). Extraction was performed by sequential solvent extraction, using different solvents of increasing polarity. Bioactivity (synergism) was monitored against the cancer cell line, and subjected to column chromatography, followed by HPLC. TLC (thin-layer chromatography) was performed to check the eluted fractions, and similar fractions were pooled together (Figure 2A). HPLC purification yielded a single sharp peak at 17.425 min retention time (Figure 2B), and further ^1^H NMR (Figure 2C), ^13^C NMR (Figure 2D), FTIR (Appendix A Appendix A) and mass spectra were analyzed (Figure 2E), to identify the identity of the compound. 

### 2.2. Chemical Characterization

The bioactive compound was isolated as a white amorphous powder. The molecular formula was deduced to be C_25_H_30_O_6_, mp.pt = 101–103 degree Celsius. IR: 2935.76, 2858.60, 1606.76, 1516.10, 1462.09, 1390.72, 1265.35, 1232.55, 1138.04, 1080.17, 1031.95, 993.37, 850.64, 815.92, 736.83 cm^−1^. Mass: Observed- 426.16, Calc:426.2051, *m/z*: 426 (6.69), 359 (23.37), 358 (100.00), 205 (22.06), 180 (16.21), 163 (33.18), 152 (26.72), 151 (95.50), 150 (19.02), 137 (50.56), 69 (18.27), 44 (6.87), 43 (9.11), 41 (9.69). ^13^C: ^δ^C: 133.64 (C-1), 108.34 (C-2), 149.68 (C-3), 146.42 (C-4), 114.23 (C-5), 118.38 (C-6), 87.71 (C-7), 54.43 (C-8), 71.03 (C-9), 130.35 (C-1′), 109.33 (C-2′), 147.96 (C-3′), 144.60(C-4′), 112.88 (C-5′), 118.41 (C-6′), 82.11 (C-7′), 50.15 (C-8′), 69.73 (C-9′), 65.82 (C-1″), 119.96 (C-2″), 137.59 (C-3″), 18.23 (C-4″), 25.83 (C-5″), 55.92 (C-3-OCH_3_), 55.99 (C-3′-OCH_3_). H-NMR, ^δ^H: 1.73(s, CH_3_), 1.77 (s, CH_3_) 2.94(m,1H), 3.37 (m,2H), 3.88 (s, OCH_3_), 3.92 (s, OCH_3_), 4.17(m,2H), 4.45 (d,1H), 4.58 (d,2H), 4.87 (d,1H), 5.53 (d,2H), 5.60 (t,1H), 6.96–6.78 (6H, Aromatic) (Appendix A Appendix A). On the basis of above data, the bioactive compound was determined to be planispine A; IUPAC name: (3R,3aR,6S,6aR)-3-(3,4-dimethoxyphenyl)-6-[3-methoxy-4-(3-methylbut-2-enoxy)phenyl]-1,3,3a,4,6,6a-hexahydrofuro [3,4-c]furan. The chemical structure of the compounds was shown in Figure 2F.

### 2.3. Planispine A Showed Synergistic Interaction with Cisplatin

To determine the effect of planispine A on cell viability and its synergistic effect with cisplatin, HeLa cells were pre-treated either with DMSO or with various doses of planispine A (2–40 µM), for 16 h. The cells were then treated with cisplatin (25 μM) for 24 h, and the viable cells were measured using a CellTiter 96 AQueous assay, following the manufacture’s protocol (Figure 3A). As shown in Figure 3B, planispine A alone can inhibit cell proliferation in a dose-dependent manner. However, a combination of planispine A and cisplatin showed a significantly increased inhibition of cell proliferation when compared with individual drug treatment. The IC_50_ of planispine A was found to be 21.38 µM and the combination treatment with cisplatin (25 µM) lowered the IC_50_ to 3.9 µM (fold change: 5.54) (*p* < 0.0001; Figure 3C). Similarly, when cells were pre-treated with 5 μM of planispine A, followed by various doses of cisplatin (6.25–100 µM) for 24 h, the IC_50_ of cisplatin were lowered from 55.3 µM to 17.43 µM (fold change: 3.17) (*p* < 0.0001; Figure 3D,E). Most of the cells were rounded up and detached from the surface when pre-treated with planispine A allowed by cisplatin. Cells which were either treated with planispine A or cisplatin showed no significant effect (Figure 3F). These results suggested that planispine A enhanced the anti-proliferation effect of cisplatin in a dose-dependent manner.

To further investigate the nature of the combination effects of planispine A and cisplatin, the Q value was determined, according to Jin’s formula, based on the cell viability assay (Figure 3G). The combination of planispine A with cisplatin showed a Q value more than one, which suggests that there is a synergistic effect when planispine A combines with cisplatin (Table 1). Moreover, the combination treatment decreases the cell proliferation compared with cisplatin alone treatment (*p* < 0.0001; Figure 3G).

### 2.4. Planispine A Enhanced Cisplati-Induced Apoptotic Cell Death

To investigate the nature of molecular interaction between planispine A and cisplatin, cells were treated either with cisplatin or planispine A, alone or in combination, as described elsewhere. Cells were stained with DAPI and analyzed using fluorescence microscopy, to observe the apoptotic nuclei. The amount of apoptotic nuclei formation was significantly higher (*p* < 0.001) in combination treatment as compared to individual treatment, indicating that combination treatment induced more efficient apoptosis (Figure 4A,B). Immunoblotting analysis showed increased caspase 3 and PARP cleavage in the combination treatment, compared with the individual treatment (Figure 4C). Cell cycle analysis showed a massive S-phase cell cycle arrest in the combination treatment, while at these low doses, the individual treatment showed no change in the cell cycle (Figure 5A). Moreover, there is reduced expression of the survival protein cyclin D1, in combination treatment (Figure 5B). Together, these data suggest that planispine A enhanced the apoptosis-inducing potential of cisplatin by modulating the apoptosis pathway.

### 2.5. Planispine A Inhibits the FA/BRCA Pathway

Since inhibition of the FA pathway sensitized the cancer cells to cisplatin, we wanted to investigate whether planispine A can inhibit the FA/BRCA pathway, which would explain the molecular mechanism of chemo-sensitization of cisplatin by planispine A. We used the FANCD2-specific antibody to visualize FANCD2 foci formation through immunofluorescence and FANCD2 monoubiquitination by immunoblotting. As shown in Figure 6A, cisplatin treatment induced FANCD2 foci formation in almost all the cells, while pre-treatment with planispine A diminished the cisplatin-induced FANCD2 foci formation (*p* < 0.001; Figure 6A,B). Similarly, immunoblotting analysis shows that cisplatin treatment induced FANCD2 monoubiquitination. However, pre-treatment with planispine A for 16 h before cisplatin treatment diminished the cisplatin-induced monoubiquitination of the FANCD2 (Figure 6C). A densitometry comparison showed that there is a 74% decrease in cisplatin-induced FANCD2 ubiquitination when pre-treated with 5 µM planispine A, and that FANCD2 monoubiquitination was totally abrogated when pre-treated with 10 µM planispine A (Figure 6C, lane 4 and 6). Together, these data suggests that planispine A can inhibit the FA pathway, which could be one of the reason for the synergistic interaction of planispine A with cisplatin.

## 3. Discussion

Cisplatin-based combination chemotherapy has been widely used in the treatment of advanced cancer, but its clinical application is limited by its toxicity and side effects, and the development of drug resistance [23,24]. Previously, *Z.armatum* has been reported to have anticancer properties and to have enhanced the therapeutic potential of cisplatin [22]. In this study, we employed a bioassay-guided purification of phytocompounds that led to the identification of planispine A (lignans) as the bioactive compound that enhanced cisplatin-induced apoptotic cell death. There has been a report of planispine A having cytotoxic potential against cancer cells [25], but no data on synergism or combination treatment has been reported up to the present date. In this study, we showed that planispine A inhibited cell proliferation in a dose-dependent manner, and IC_50_ was found to be 21.64 µM which was further lowered to 3.9 µM when combined with cisplatin (fold change: 5.54 times). Similarly, IC_50_ for cisplatin was lowered from 55.3 µM to 17.43 µM when combined with planispine A. The fold change of the combination treatment was found to be 3.17 times. Moreover, combined treatment increases the efficacy of cisplatin-induced apoptosis in cancer cells in a caspase-dependent manner. Problems with cisplatin treatment include drug resistance and toxicity due to high doses of chemotherapeutic drugs [6,26,27]. Our data on the planispine A-cisplatin combination indicates that a novel combination can be used as a chemo-sensitizer that may sensitize the drug-resistant cancer cells with minimal toxicity, due to the use of a low dose of cisplatin with high efficacy.

Increased activation of the DNA repair pathway is one of the reasons for the chemo-resistance of cisplatin. The Fanconi anemia (FA) pathway was designed to repair DNA damage induced by ICL, and involved twenty two genes [28]. Eight of them formed a core complex, and the remaining formed the downstream proteins [29]. The mutation of core complex proteins inhibits FANCD2 monoubiquitination and FANCD2 foci formation, while the mutation of downstream proteins has no effect on monoubiquitination but compromises FANCD2 foci formation [30]. The mechanisms of FA pathway inhibition by planispine A remain unknown. Since planispine A inhibits both monoubiquitination and foci formation, it is likely that planispine A may inhibit the upstream pathway. Whether planispine A directly targets the components of the FA pathway remains to be determined, and further studies of the molecular targets affected by planispine A are warranted.

Planispine A and cisplatin showed higher apoptosis of cancer cells, suggesting that these combinations may be relevant for cancer treatment, and may have therapeutic advantages in the treatment of cisplatin-resistant cancers. Further analysis of this combination or its analogs may reveal more specific and less toxic drugs, which would need pre-clinical animal models.

## 4. Materials and Methods

### 4.1. Plant Material Extraction

The specimen (*Zanthoxylum armatum* DC leaf) was collected from Imphal-East, Manipur, India at the coordinates 24.7807° N and 93.9674° E, and was identified by a taxonomist of the Plant Systematic and Conservation Lab, Institute of Bioresources and Sustainable Development (IBSD), Manipur. A voucher specimen (IBSD/M-223/2017) has been deposited at the herbarium. The leaves were collected, washed, dried at room temperature, and grounded using an electric blender. The powdered samples (1.43 kg) were macerated in methanol for 2–3 days, filtered, and concentrated using a rotary vacuum evaporator (Rotavapor R-3, Buchi, Switzerland). Methanol was completely removed and kept at −20 °C until further analysis.

### 4.2. Bioassa-Guided Purification of the Bioactive Compounds

The crude methanolic extract (100 g, dark brown) was subjected to column chromatography using silica gel (200–400 mess) with successive elution by different solvents of increasing polarity such as hexane, ethyl acetate, butanol, methanol and water, yielding five fractions as shown in Figure 1. The bioactivity tests for synergism with cisplatin were monitored against the cancer cell line, and the ethyl acetate fractions were found to show activity (15.16 g, dark green). They were subjected to further purification by silica column chromatography, and elutions were performed using a hexane and ethyl acetate gradient, yielding eleven fractions. The eluted fractions were checked using TLC (thin-layer chromatography) (Merck, Kieselgel 60 F254 0.25 mm), and similar fractions were pooled together, followed by the removal of solvents using a rotary evaporator. The bioactivity was again monitored, and fractions with hexane/ethyl acetate (5:5) were found to be active (5.38 g, light green). The active fractions were further purified by preparative HPLC (Shimadzu, LC-20AP), using a reverse-phase C-18 column (10 × 250 mm i.d, 5 µm). The samples were dissolved in methanol and filtered through a 0.2 μm syringe filter, prior to injection onto the column. The detector wavelength was set at 280 nm, and solvent flow rate at 2.0 mL/min. Elution began with 70% MeOH/H_2_O, isocratic for 3 min, then a linear gradient of 70% to 100% MeOH at 25 min. Each peak was collected and tested for its bioactivity. One of the purified compounds showed synergistic interaction when treated in combination with cisplatin, to yield a single sharp peak at 17.425 min retention time. ^1^H, ^13^C NMR and mass spectrometry were carried out at Tokushima Bunri University, Tokushima Japan. The Samples were dissolved in DMSO and kept at −20 °C until further use.

### 4.3. Cell Culture

Hela cells were purchased from the National Centre for Cell Science (NCCS) Pune, India. The cells were propagated in DMEM medium, supplemented with 10% fetal bovine serum, 100 U/mL Pen-Strep (GIBCO, Life Technologies, Carlsbad, CA, USA). The cells were incubated in a humidified atmosphere with 5% CO_2_ at 37 °C.

### 4.4. Cell Viability Assay

Cell viability was measured using CellTiter 96 AQueous One Solution Reagent (Promega), following the manufacturer’s protocol. The cells (5 × 10^3^ cells/well) were seeded in 96-well tissue culture plates, incubated overnight to adhere the cells, and then pre-treated with planispine A for 16 h prior to the treatment with cisplatin for 24 h. Subsequently, 20 μL of CellTiter reagent was added to each well, and the plates were incubated for 1–4 h. The absorbance was measured at 490 nm on a microplate reader (Multiskan Go, Thermo Fisher Scientific, Waltham, MA, USA). The untreated cells were used as a control of viability (100%) and the results are expressed as % viability relative to the control. All the experiments were conducted three times, in triplicate.

### 4.5. Drug Combination Effect Analysis

The combination effect of planispine A and cisplatin was evaluated by Jin’s formula; Q = Ea + b/(Ea + Eb − Ea × Eb) where Ea + b, Ea, and Eb are the inhibition rates of the combination group, drug a, and drug b, respectively. Q = 1 indicates addition; Q > l, synergism, Q < 1, antagonism [31,32].

### 4.6. Apoptosis Assay by Immunofluorescent Staining

Cells were seeded into 6-well tissue culture plates (5 × 10^4^ /well) containing sterile 22 × 22 mm coverslips for 24 h, to allow adherence to coverslips. After treatment, the coverslips were washed with phosphate buffer saline (PBS), permeabilized with 0.5% triton X-100, and fixed with 4% paraformaldehyde. The cells were washed, stained with 4,6-diamidino-2-phenylindole, DAPI (Invitrogen), and apoptotic nuclei were observed with a fluorescent microscope with appropriate filters (Leica DMi8 Microscope, GmbH, Wetzlar, Germany).

### 4.7. Cell Cycle Analysis

Cells were seeded on 6-well tissue culture plates (5 × 10^4^ /well) overnight, followed by treatment with the indicated planispine A or cisplatin alone or a combination, for 24 h. The cells were harvested and fixed with 70% ethanol for 30 min at 4 °C. The cells were stained with propidium iodide (50 μg/mL) in the presence of 100 ug/mL RNase A, and cell cycles were determined using flow cytometry (BD Acuri C5).

### 4.8. Western Blots

Cells were seeded into 60 mm tissue culture plates (3 × 10^6^/well), and pre-treated with planispine A (5 and 10 μM) for 16 h, prior to the treatment with cisplatin (50 μM) alone or in combination, for 24 h. The cells were harvested, lysed, and equal amount of proteins were separated on SDS-PAGE, and then transferred onto PVDF membranes. The membranes were blocked and probed with primary antibodies against anti-cleaved PARP, anti-caspase 3, anti-cyclin D1, anti-actin and anti-FANCD2 primary antibodies. All antibodies were procured from Cell Signalling, MA, USA. The membranes were washed, incubated with secondary antibodies, and the protein bands of interest were visualized using ChemiDoc Imaging System (Bio-Rad, Hercules, CA, USA).

### 4.9. Confocal Microscopy

Confocal microscopy was performed as described previously [33]. Cells were grown on coverslips coated with poly-lysine overnight and treated with planispine A or cisplatin alone or a combination, for 24 h. The adherent cells were washed with PBS, and fixed with 4% paraformaldehyde in PBS for 10 min. The cells were permeabilized with 0.5% Triton X-100 in PBS for 5 min, and blocked with 1% bovine serum albumin in PBS for 1 h. The fixed cells were incubated overnight with anti-FANCD2, washed in PBS supplemented with 0.2% Tween 20 (PBST), and incubated with FITC-labelled secondary antibodies for 45 min. The cells were washed with PBST, stained with DAPI, mounted on glass slides and observed with a Nikon Eclipse Ti2 inverted microscope.

### 4.10. General Experimental Procedures

IR spectra were measured on a Shimazu FTIR-8400S instrument. Proton NMR. 13C and 2-dimensional NMR spectra were obtained on a Varian UNITY 600 NMR spectrometer. The chemical shifts are given in G (ppm), and coupling constants are reported in Hz. Mass (HRMS) spectra were obtained on a JEOL JMS-700 instrument. Kieselgel 60 (230–400 mesh, Merck, Kenilworth, NJ, USA) was used for column chromatography, and silica gel 60 F-254 (Merk) for TLC. HPLC analysis was performed on a SHIMADZU LC-20AP 02027/02028 pump coupled to a SHIMADZU SPD-20A 40766 UV–vis detector set at 280 nm, using a reverse-phase C-18 column (10 × 250 mm i.d, 5 µm).

### 4.11. Statistical Analysis

All the experimental data were represented as mean ± SD of three independent experiments. Statistical analysis was performed using one-way analysis of variance (ANOVA) and Tukey- Kramer multiple comparison tests using GraphPad Prism 8.0. The *p* value < 0.05 was considered statistically significant.

## 5. Conclusions

In conclusion, this study underscores the potential clinical benefit of combination therapy using cisplatin and planispine A.

## Figures and Tables

**Figure 1 molecules-27-07288-f001:**
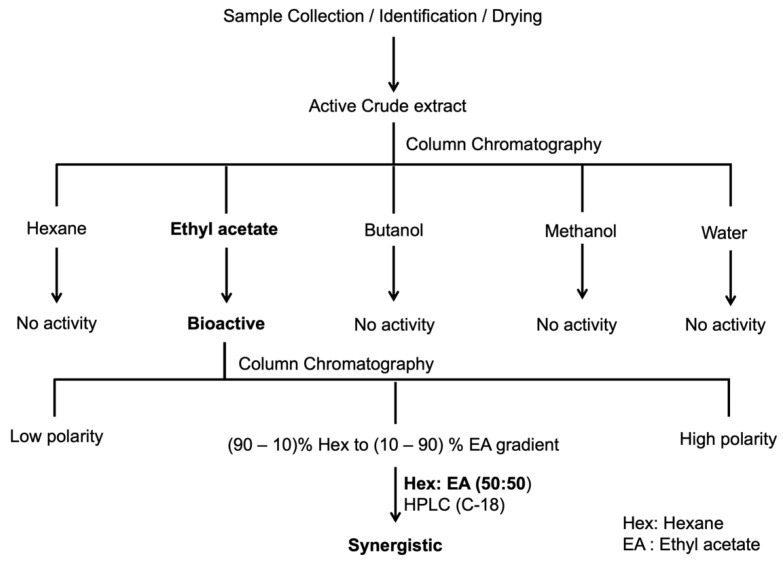
Flowchart depicting process of extraction and bioassay-guided purification of *Zanthoxylum armatum* DC leaf extracts. Successive extractions using different solvents of increasing polarity (hexane < ethyl acetate < butanol < methanol < water) to purify bioactive fractions. The extracts were subjected to fractionation using silica column chromatography and thin-layer chromatography (TLC).

**Figure 2 molecules-27-07288-f002:**
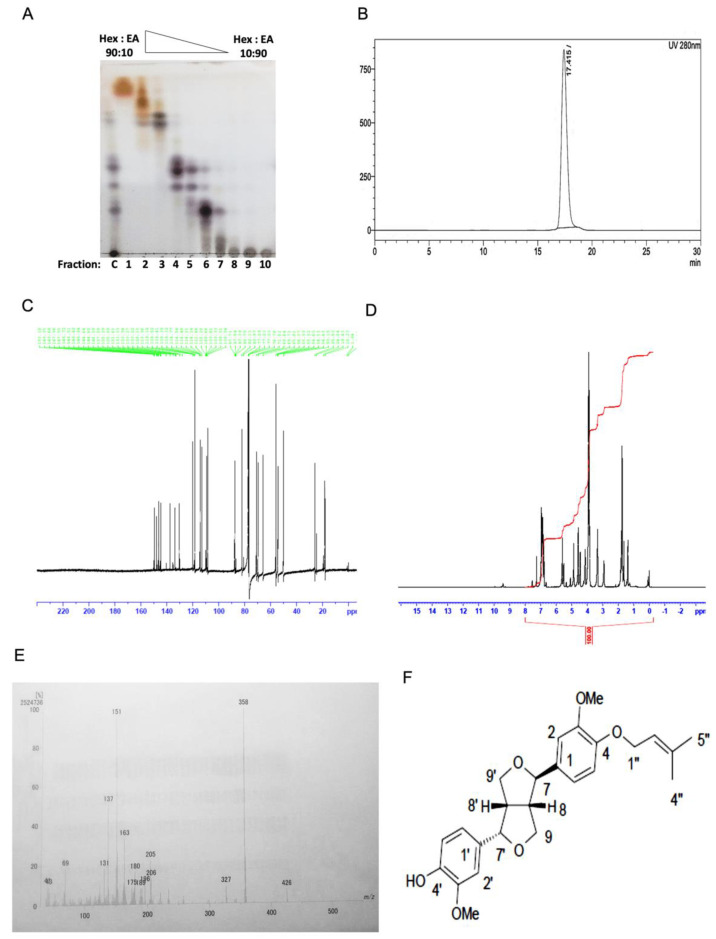
Characterization of bioactive compounds. (**A**) TLC chromatograms of bioassay-guided purification from silica gel column chromatography. Eluent of (90–10) % Hexane to (10–90) % Ethyl acetate gradient were used to elute the fractions. The eluted fractions (1–10) and crude methanol extract (**C**) were analyzed by TLC, using 60 F254 silica gel at room temperature. TLC was run at Hexane: Ethyl acetate (1:1, *v/v*) and detected by sulphuric acid spotting test. (**B**) HPLC chromatograms of isolated and purified compound: HPLC analysis was performed on a SHIMADZU LC-20AP 02027/02028 pump coupled to a SHIMADZU SPD-20A 40766 UV–vis detector, set at 280 nm. (**C**–**E**) ^13^C NMR, ^1^H-NMR, and mass spectrometry spectra of the purified compound as described elsewhere (**F**) Chemical structure of planispine A.

**Figure 3 molecules-27-07288-f003:**
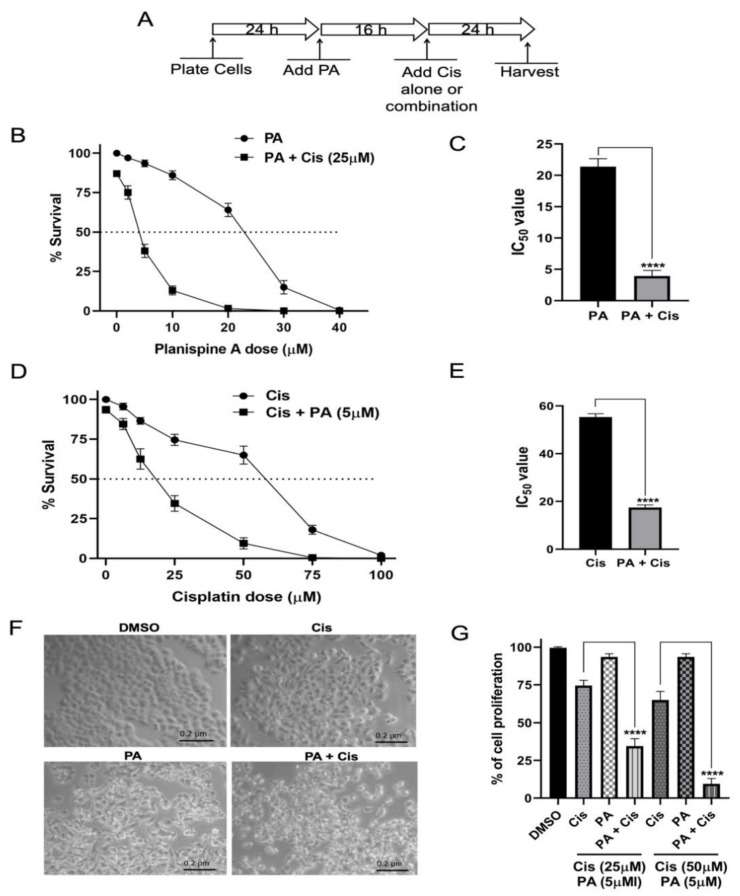
Planispine A acts in synergism with cisplatin. (**A**) Schematic protocol for the combination treatment. (**B**) Dose-dependent inhibition of cell proliferation by planispine A or in combination with cisplatin (25 μM). (**C**) Graph representing IC_50_. (**D**) Dose-dependent inhibition of cell proliferation by cisplatin or in combination with planispine A (5 μM). (**E**) Graph representing IC_50_ values. (**F**) Microscopy image showing morphology of cells when treated alone or combination with planispine A and cisplatin (20X). Scale bar, 0.2 μm. Cells were treated with DMSO (control), cis (cisplatin, 25 µM), PA (planispine A, 5 µM) or a combination. (**G**) Quantification of cell proliferation assay. Cells were pre-treated with PA (5 µM) for 16 h before treatment with cis (25 and 50 μM) for 64 h. Data represent percentage of growth compared with DMSO. **** *p* < 0.0001 vs. cisplatin.

**Figure 4 molecules-27-07288-f004:**
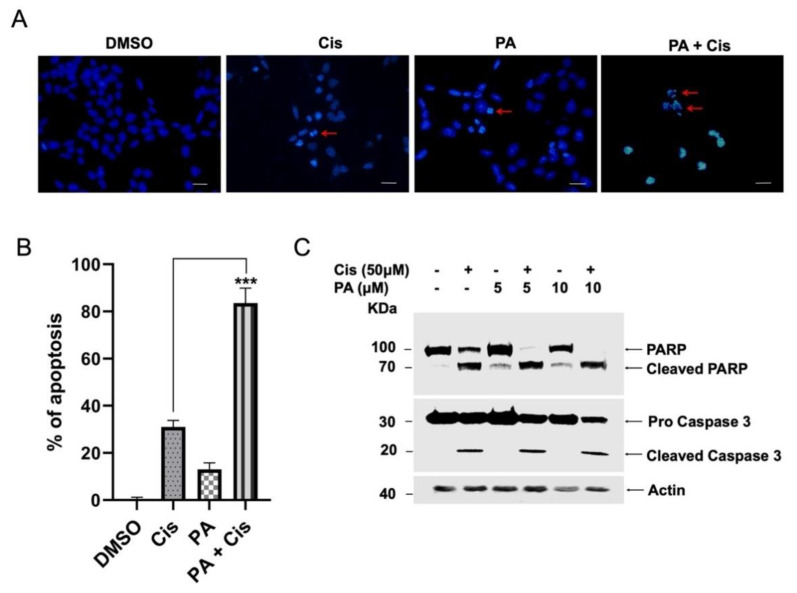
Planispine A-enhanced cisplatin induced apoptosis. (**A**) Fluorescence microscopy images of DAPI-stained cells treated with DMSO (control), cisplatin; cis (50 µM), planispine A; PA (5 µM) and combination treatment for 64 h. The red arrows indicate the apoptotic fragmented nuclei (40X). Scale bar, 5 μm. (**B**) Representative quantification graph for apoptotic nuclei. Data are representative of three independent experiments, *n* = 3, *** *p* < 0.001 vs. cisplatin. (**C**) Immunoblot with anti-PARP and anti-caspase-3 after treatment as indicated.

**Figure 5 molecules-27-07288-f005:**
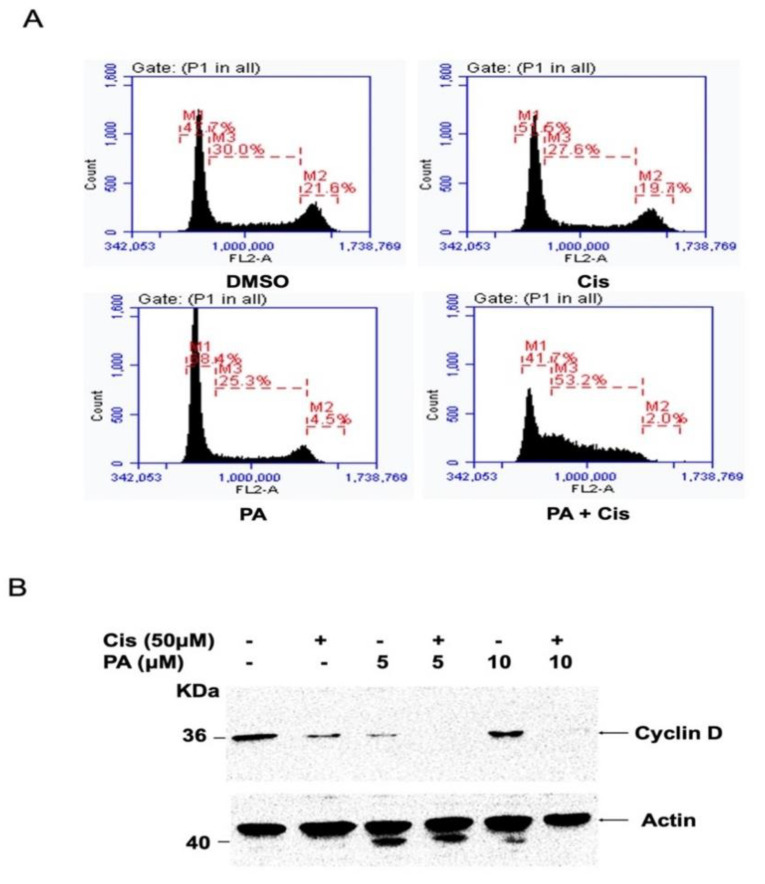
Combination of planispine A and cisplatin leads to S-phase arrest. (**A**) Cells were treated with DMSO (control), cisplatin; cis (2 µM), planispine A; PA (3 µM) and a combination. Cells were fixed with 70% ethanol and stained with Propidium iodide and cell cycle was determined by flow cytometry. (**B**) Immunoblot showing expressions of cyclin D1 after treatment as indicated.

**Figure 6 molecules-27-07288-f006:**
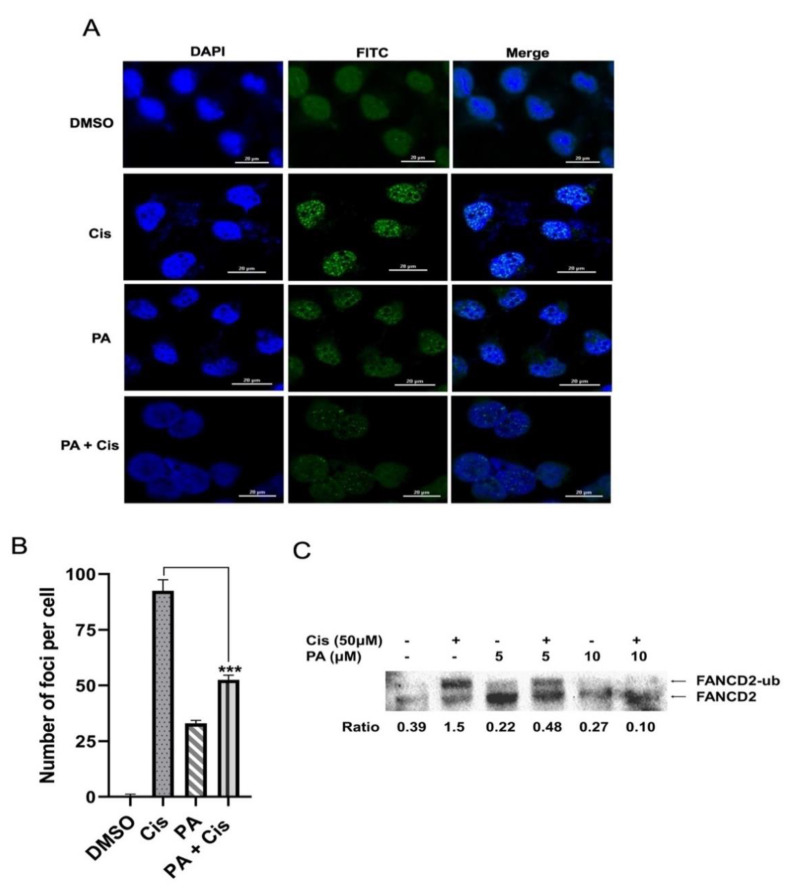
Planispine A inhibits the FA/BRCA pathway. (**A**) Immunofluorescence for FANCD2 foci; percentage of cells with more than 5 foci were determined in at least 150 cells (60X). Scale bar, 20 μm. (**B**) Graphical representation FANCD2 foci, *n* = 3, *** *p* < 0.001 vs. cisplatin. (**C**) Western blot confirming the inhibition of FANCD2 monoubiquitination by pre-treatment with planispine A. Densitometry analysis to measure the ratio of monoubiquitination FANCD2 to non-ubiquitinated FANCD2 is shown below the blot. “+” indicates in presence of drug and “–” indicates without drug.

**Table 1 molecules-27-07288-t001:** Combination index of planispine A and cisplatin using Jin’s formula. The data are expressed as the mean ± S.D. in triplicate. (*p* < 0.01).

Drugs	% Inhibition Rate	Q Value
		(Combination Index)
PA (5 µM)	7.4 ± 2.08	-
Cisplatin (25 µM)	24 ± 3.6	-
Cisplatin (50 µM)	35 ± 4	-
PA (5 µM) + Cisplatin (25 µM)	65.4 ± 3.5	2.23
PA (5 µM) + Cisplatin (50 µM)	89.4 ± 1.5	2.44

Q = 1 (addition); Q > l (synergism); Q < 1 (antagonism).

## Data Availability

All data used to support the findings of this study are included within the article and the Appendix A.

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
