# Peer review of "Planispine A Sensitized Cancer Cells to Cisplatin by Inhibiting the Fanconi Anemia Pathway"

_molecules, 2022, doi:10.3390/molecules27217288_

Round 1

Reviewer 1 Report

This manuscript explores the significance of Planispine A usage in sensitizing cancer cells to cisplatin by inhibiting the Fanconi anemia pathway. Overall, the manuscript is well written, and the presented results favour the conclusion. Nevertheless, quantitative PCR or real-time polymerase chain reaction would be a handy tool in such a project for comprehensive results. Moreover, wound healing assay is another recommended technique for probing collective cell migration as a comparison study before and after the different treatments. These aspects will add further accuracy to the results; nevertheless, the current contents are satisfactory enough for publication. 

The materials and methods section shows enough information. Graphs and figures are clear enough to build the discussion and draw a conclusion.

Author Response

Comments and Suggestions for Authors 1

This manuscript explores the significance of Planispine A usage in sensitizing cancer cells to cisplatin by inhibiting the Fanconi anemia pathway. Overall, the manuscript is well written, and the presented results favour the conclusion. Nevertheless,

Comments: quantitative PCR or real-time polymerase chain reaction would be a handy tool in such a project for comprehensive results.

Our reply: We agree with your suggestion that qPCR would have strengthen our results. But since the protein quantity rather than the mRNA is more relevant with regards to the physiological relevant, we have chosen to detect protein only.

Comments: Moreover, wound healing assay is another recommended technique for probing collective cell migration as a comparison study before and after the different treatments.

Our reply: We totally agree that migration study is important, but it is beyond the scope of this study. We plan to study migration and EMT in near future.

Comments: These aspects will add further accuracy to the results; nevertheless, the current contents are satisfactory enough for publication. The materials and methods section shows enough information. Graphs and figures are clear enough to build the discussion and draw a conclusion.

Our reply: Thank you

Reviewer 2 Report

The manuscript: Planispine A sensitized cancer cells to cisplatin by inhibiting the Fanconi anemia pathway by T.R. Singh and collaborators is of great interest in the cancer treatment of future patients.

I have a few points: Could this combination be used for other anti-cancer agents such as Carboplatin or unrelated anti cancer agent related to Taxol derivatives. Are any data available in the Lit?

Are there any changes to the development of resistance towards cis-Platin observed?

These points could be added to the Discussion in the manuscript if available.

Author Response

Comments and Suggestions for Authors 2

The manuscript: Planispine A sensitized cancer cells to cisplatin by inhibiting the Fanconi anemia pathway by T.R. Singh and collaborators is of great interest in the cancer treatment of future patients.

Comments: I have a few points: Could this combination be used for other anti-cancer agents such as Carboplatin or unrelated anti cancer agent related to Taxol derivatives. Are any data available in the Lit?

Our reply: We have not investigated the combination of planispine A and Carboplatin. However, since the mode of action of cisplatin and Carboplatin are the same, we expect a similar synergistic interaction. We didn’t also study the combination with Taxol and will surely check in our later study.

Comments: Are there any changes to the development of resistance towards cis-Platin observed? These points could be added to the Discussion in the manuscript if available.

Our reply: We did not see development of cisplatin resistance and rather we are expecting sensitization of cisplatin resistant cells which is something we will explore later. We thank you for your suggestions.

Reviewer 3 Report

The manuscript submitted by Singh et al. describes a new compound that can act synergistically with cisplatin and therefore, may help to overcome drug resistance. It is scientifically sound, the experiments are well designed, and the results clearly explained. However, some minor corrections, detailed below, must be made before its acceptance for publication in Biomolecules.

1.    Comment on the text:

§  Why did the addition of cisplatin or planispine A (PA) alone not have an effect on cell cycle, despite their effect on cell viability and on the amount of cyclin D?

§  Why was the amount of cyclin D higher when the cells are treated with PA 10µM than with 5µM? See figure 5B, lanes 3 and 5.

2.    Define the abbreviations used in the text: MMC and TLC

3.    Correct the enumeration of the panels in Figure 2 in the first paragraph of the Results section.

4.    In section 2.3, first paragraph, substitute combination by synergistic

5.    Provide a more detailed explanation of the techniques enumerated in 4.10. Include in this section mass spectrometry, which is missing.

6.    In general, the English language must be revised. Some prepositions, articles, and commas are missing throughout the document. Some verbal tenses are not correct.

Author Response

Comments and Suggestions for Author 3

The manuscript submitted by Singh et al. describes a new compound that can act synergistically with cisplatin and therefore, may help to overcome drug resistance. It is scientifically sound, the experiments are well designed, and the results clearly explained. However, some minor corrections, detailed below, must be made before its acceptance for publication in Biomolecules.

  1. Comment on the text:

Comments: Why did the addition of cisplatin or planispine A (PA) alone not have an effect on cell cycle, despite their effect on cell viability and on the amount of cyclin D?

Our reply: We have chosen very low doses of cisplatin and PA (2µM and 3µM respectively) where there is no cell death. we thank the reviewer for asking this question because we missed out to mention the doses. Accordingly, we have added the doses at the fig. 5A legend.

Comments: Why was the amount of cyclin D higher when the cells are treated with PA 10µM than with 5µM? See figure 5B, lanes 3 and 5.

Our reply: We appreciate your important observation. We think that at low dose (5µM), there is no appreciable cell death. However, when the dose is increased to 10µM (IC50 is around 21µM), cells has just started dying. It is possible that cells started activating survival signals.

Your comment: Define the abbreviations used in the text: MMC and TLC

Our reply: We thank you for bringing out these mistakes. Accordingly, MMC and TLC abbreviations have been defined at para 4 of “Introduction” section and section 2.1.

Comments: Correct the enumeration of the panels in Figure 2 in the first paragraph of the Results section.

Our reply: The required changes have been made as suggested.

Comments: In section 2.3, first paragraph, substitute combination by synergistic

Our reply: As suggested combination was replaced by synergistic.

CommentsProvide a more detailed explanation of the techniques enumerated in 4.10. Include in this section mass spectrometry, which is missing.

Our reply: Mass spectrometry details are mentioned in 4.10

Comment:  In general, the English language must be revised. Some prepositions, articles, and commas are missing throughout the document. Some verbal tenses are not correct.

Our reply: As advised we have revised the English to our level best.

Reviewer 4 Report

The manuscript by Singh et al. presents the identification of a compound from the plant extracts of Zanthoxylum armatum DC. Based on their previous results showing that the Zanthoxylum armatum DC extract showed synergistic anti-cancer activity with cisplatin. The authors went distances to identify and purify the compound (Planispine A) and characterized the activity of this compound in cell-based assays as well as synergy with cisplatin. This is excellent work considering the methodology involving the purification of the compound. There are however some concerns that should be addressed prior to the publication. 

1. The figure quality of all figures is poor. I am not sure if it is due to the incorporation of figures in the WORD document or they were of poor quality, to begin with.

2. The labels on Fig. 2C are literally invisible. There are no labels on Fig. 2A. It will be a good idea to include a chemical (IUPAC) name for the compound.

3. The IC50 is not a range! It is a value and that is how it should be presented. Also, what does "more significant" mean? There is no such thing as a potency ratio. It is called fold-change, and that is how it should be written.

4. Expand on conclusions to include/hypothesize a mechanism of synergy with cisplatin.

5. What is MMC? Please explain.

Author Response

Comments and Suggestions for Authors 4

The manuscript by Singh et al. presents the identification of a compound from the plant extracts of Zanthoxylum armatum DC. Based on their previous results showing that the Zanthoxylum armatum DC extract showed synergistic anti-cancer activity with cisplatin. The authors went distances to identify and purify the compound (Planispine A) and characterized the activity of this compound in cell-based assays as well as synergy with cisplatin. This is excellent work considering the methodology involving the purification of the compound. There are however some concerns that should be addressed prior to the publication. 

  1. Your comment: The figure quality of all figures is poor. I am not sure if it is due to the incorporation of figures in the WORD document or they were of poor quality, to begin with.

Our reply: We agree that figure quality may be poor and it is due to the incorporation of the figures inside the text. High resolution ( atleast 300dpi) are also available which can be used.

  1. Your comment: The labels on Fig. 2C are literally invisible. There are no labels on Fig. 2A. It will be a good idea to include a chemical (IUPAC) name for the compound.

Our reply: The invisibility in Fig. 2C is due to resizing the figure. We have provided the original figure as supplementary number S4. Fig. 2A has been labelled. The IUPAC name of the compound has been inserted at section 2.2.

  1. Your comment: The IC50 is not a range! It is a value and that is how it should be presented. Also, what does "more significant" mean? There is no such thing as a potency ratio. It is called fold-change, and that is how it should be written.

Our reply: As suggested, range in IC50 has been removed and only the mean IC50 is reported. Potency ration is also replaced with fold-change. “More significant” was replaced with “significant”

  1. Your comment: Expand on conclusions to include/hypothesize a mechanism of synergy with cisplatin.

Our reply: We have included the possible mechanism of synergism in the discussion section. We hypothesized that planispine A inhibit FA pathway that is responsible for the repair and removal of cisplatin crosslinking the DNA.

  1. Your comment: What is MMC? Please explain.

Our reply: MMC is Mitomycin C and as suggested it has been explained in para 4 of “Introduction” section.